# Association of Performance in Strength and Plyometric Tests with Change of Direction Performance in Young Female Team-Sport Athletes

**DOI:** 10.3390/jfmk6040083

**Published:** 2021-10-14

**Authors:** Hallvard Nygaard Falch, Eirik Lindset Kristiansen, Markus Estifanos Haugen, Roland van den Tillaar

**Affiliations:** Department of Sport Sciences and Physical Education, Nord University, 7600 Levanger, Norway; falch7@hotmail.com (H.N.F.); ek1105@hotmail.com (E.L.K.); haugen@helseogprestasjon.no (M.E.H.)

**Keywords:** force, velocity, power, step kinematics

## Abstract

The change of direction (COD) ability is a task-specific skill dependent on different factors such as the degree of the turn, which has led to differentiating CODs as more force- (>90°) or velocity-oriented (<90°). Considering force and velocity requirements is of importance when designing sport-specific training programs for enhancing COD performance. Thus, 25 female handball and soccer players participated in this study, which investigated the association between three different strength and plyometric exercises and force- and velocity-oriented COD performance. By utilizing the median split analysis, the participants were further divided into a fast (*n* = 8) and a slow (*n* = 8) COD group, to investigate differences in step kinematics between fast and slow performers. The correlational analysis revealed that the bilateral back squat and unilateral quarter squat were significantly associated with several force- and velocity-oriented COD performance (*r* = −0.46 to −0.64), while the association between plyometric and COD performance was limited (*r <* 0.44). The fast COD group revealed higher levels of strength, jump height, peak velocities, higher step frequencies, shorter ground contact times, and greater acceleration and braking power (*d* > 1.29, *p* < 0.03). It was concluded that the observed correlation between strength and COD performance might be due to stronger athletes being able to produce more workload in a shorter time, which was supported by the step kinematics.

## 1. Introduction

In court and field sports, athletes are required to possess numerous physical and tactical skills [1,2]. The physical skills required are distinctive for the different sports, positions on the field, and team tactics [3,4]. However, a minimum threshold for maximal oxygen consumption to endure the match is required [5,6], accompanied by the ability to perform and repeat various high-intensity actions throughout the match [7,8,9]. Despite the short timeframe of these high-intensity actions relative to total match-time, they are decisive for match outcomes as they often precede match-decisive situations [10,11]. One such important high-intensity action is the physical ability to rapidly perform a change of direction (COD), which frequently occurs in team sports [12,13,14]. The COD ability can be defined as the athlete’s preplanned physical ability to accelerate and decelerate while overcoming inertia before reaccelerating in a new direction [15,16,17]. Athletes with a well-developed COD ability possess a physical and tactical advantage over their opponent, as, in offensive play, they may have an increased opportunity of bypassing their opponent, creating space, and/or moving into space. Generally, lower-limb strength, power, and reactive strength are leg muscle qualities positive for COD performance [18], due to the importance of rapidly expressing a large amount of force over multiple steps when performing a COD [16,19,20].

Although leg muscle qualities are important factors determining COD performance in general, COD is a task-specific skill [21,22], as improvement in one specific COD task might not transfer to another [18,23]. Factors constraining the specifics of a COD maneuver can be the complexity (i.e., handling a ball) [21], number of COD(s) performed [15], completion time (energy systems utilized), initial velocity approaching the COD, and angle of the turn [19,21,24]. CODs performed at a higher initial velocity will increase braking requirements to manage the turn [19], as shown by more steps for braking [22]. Furthermore, CODs of greater angles require greater redirections of the whole body [25], increases in ground contact time, greater mediolateral force production, and greater knee and hip flexion in the plant step [22,25,26,27,28]. Accordingly, CODs of different angles have been suggested to demand different magnitudes and directions of ground reaction forces. This concept is rooted in Newton’s laws of motion, as CODs of greater angles (>90°) require greater force productions for braking to change momentum and redirecting the body, as opposed to velocity-oriented CODs (<90°), allowing for velocity maintenance and a transfer of momentum [20,21,24].

Strength and plyometric training are two different training modalities often utilized to improve leg muscle qualities, which may furthermore lead to enhanced COD performance [18,21]. Unfortunately, the distinctiveness of different types of CODs is often neglected when testing COD performance, whereby the COD ability is processed as a general skill, without considering the specific demands of CODs, such as force and velocity magnitudes. Thus, results after conducting training interventions might conflict [21]. To the authors’ knowledge, a study by Falch et al. [29] is the only one to specifically examine the association of strength and plyometric exercises with force- and velocity-oriented CODs. The study was conducted on male soccer players and suggested that the plyometric exercises should be COD-specific, while the different strength tests revealed only a ‘moderate’ insignificant correlation (*r* < 0.41). Furthermore, a review by Falch, Rædergård, and van den Tillaar [21] suggested strength training to be more beneficial for developing force-oriented COD performance, as force-oriented CODs are performed at slower velocities. However, it is unknown how these findings apply to female athletes who are underrepresented in COD research [21]. Females possess, on average, less strength and more fat mass in comparison to males [30], which are important considerations when training for enhancing COD performance. Enhancing relative strength might increase COD performance [31], as acceleration/deceleration of the body is a product of net force [24]. Therefore, strength training might be more beneficial for female athletes for developing overall COD performance due to generally lower levels of relative strength. This assumption is reasonable, as strength training has been found to enhance COD performance in both female volleyball athletes and untrained females when measuring performance changes using a standardized *t*-test [32,33].

Although earlier research suggests plyometric training to be more COD-specific [21,23,29], relative strength might reveal a high association for female athletes in both force- and velocity-oriented CODs, as there might be a “threshold” before further strength gains will not enhance COD performance. Lower-limb strength, commonly dynamically expressed with exercises such as the barbell back squat, might be of even greater importance in force-oriented CODs, as CODs of increased angles require longer ground contact times with greater knee and hip-joint angles [22]. Due to the limited time available to devote to training specific physical abilities such as the COD ability in team sports, it is important to incorporate exercises and training modalities positively affecting overall performance outcomes in a sport-specific context. Thus, the primary objective of the current study was to examine the association between young female court and field sport athletes’ performance in different strength and plyometric exercises with performance in a force- and velocity-oriented COD. A secondary objective was to investigate differences in step kinematics and acceleration/deceleration between fast and slow performances in force- and velocity-oriented CODs for further insights into the demands of the force -and velocity-oriented CODs. It was hypothesized that fast COD performers would be better at both accelerating and decelerating in both force- and velocity-oriented CODs due to higher step frequencies and shorter contact times [34]. Such an investigation might provide useful information for strength and conditioning coaches seeking to improve task-specific COD performance in female athletes. Performance in the strength and plyometric exercises was hypothesized to be associated with performance in both force- and velocity-oriented CODs, as all performances are a product of lower-limb force production relative to body mass.

## 2. Materials and Methods

A within-subject design was conducted during the offseason to investigate the association between performance in strength and plyometric exercises and performance in force- and velocity-oriented CODs in female court and field sport athletes. A between-subject design was used to examine differences between fast and slow COD performers. To avoid a possible learning effect, all athletes had to participate in two familiarization sessions, practicing the different tests of the study. Technical guidance (mainly foot placement and depth) and the study procedure for the familiarization and test day were controlled for by three strength and conditioning professionals.

### 2.1. Subjects

The subjects of the current study consisted of 25 young female handball (*n* = 16) and soccer players (*n* = 9) (age: 19.6 ± 2.8 years, height: 170 ± 7.1 cm, body mass: 68 ± 10.6 kg, body mass index: 23.6 ± 2.3) participating in a minimum of three sessions per week, recruited from two local teams. Both teams competed at the second-highest level in the Norwegian league system of their respective sport. None of the athletes had an injury or any illness in the previous 3 months that could negatively affect the validity of the study. All athletes were informed of the risks and benefits of participation, and a written consent form from the athletes and parents (when under 18 years old) was obtained before the tests. The study was conducted according to the ethical regulations for research, approved by the Norwegian Center for Research Data (project number: 903955) in line with the latest revision of the Declaration of Helsinki. The athletes were instructed to be physically and mentally prepared for performing maximal efforts on the day of testing, which involves the consumption of a light meal 2 h before testing, >7 h of sleep, not consuming alcohol, and avoiding demanding physical training 48 h before testing.

### 2.2. Procedure

Before testing the performance in the different strength, plyometric, and COD tests, all athletes underwent the standardized warm-up protocol presented by van den Tillaar et al. [35] consisting of submaximal runs and dynamic stretching. Afterward, athletes were randomly assigned by an online randomizer to three different groups, testing maximum performance in the different tests (strength, plyometric, and COD/running). Each group was randomly assigned to start with the strength, plyometric, or COD tests and completed all the tests within the session. Because the athletes already performed a warm-up protocol, only sub-maximal repetitions of the different tests were included as a specific warm-up, leading up to the maximal effort attempts. To investigate the association of force- and velocity-oriented COD performance with strength and plyometric exercises, the athletes were required to perform one velocity- (45°) and one force-oriented (180°) COD task. The COD test was accompanied by a sprint test (large magnitude of concentric force and velocity requirements) and a braking test (large magnitude of eccentric braking force requirements) [36]. Furthermore, the athletes performed three plyometric tests and three strength tests, which were performed bilaterally and unilaterally with the right foot in the vertical and lateral directions. The right foot performed the unilateral movements as the right foot performs the plant step in CODs with a left turn [22].

### 2.3. COD Tests

The COD tests consisted of a 10 m sprint approaching a 45° or 180° turn and performing the turn before reaccelerating 10 m into the new direction. All CODs were performed with a left turn to ensure that the right foot performed the pivoting step. For the 45° COD, the athlete had to run 10 m before performing the plant step before the 0.8 m line after the 10 m line; she then proceeded by running toward the new direction. For the 180° COD, the athlete had to run 10 m, placing the pivoting foot on the 0.8 m line before reaccelerating into the new direction. Athletes were instructed to complete the test within the shortest amount of time possible. Both total time (10 m + COD + 10 m) and partial time (first and last 10 m) were measured (Figure 1).

In all COD, sprint, or braking tests, the athlete was required to start from a standstill position, with the front foot placed 5 cm behind the first timing gate to prevent a false trigger of a random limb. Time was started after passing the first wireless timing gate (Brower Timing Systems, Salt Lake City, UT, USA, height of 1 m) and stopped when passing the last timing gate (Figure 1 and Figure 2). The tests started on the athletes’ own accord, after receiving a signal from a researcher, to limit a reactive component to the different tests. A contact grid, IR-Contactmat-ML6TJP02-870 (Ergotest Innovation, Porsgrunn, Norway,) was utilized to investigate step kinematics in the COD, sprint, and braking tests. The contact grid detected contact and flight times and was placed along the starting line for the COD, sprint, and braking tests. The contact grid covered the whole sprint and braking test area for force-oriented COD, while, for velocity-oriented COD, it covered the first 10 m (Figure 1).

Distance and velocity in the COD, sprint, and braking tests were measured and calculated using a wireless CMP3 distance sensor laser gun positioned 1.8 m behind the athlete (Noptel Oy, Oulu, Finland), with a sampling at 2.56 kHz, which was pointed at the athlete’s lower back (approximately center of mass) while running.

Step kinematics were also calculated from the COD test (average step length, average strep frequency, average ground contact and flight times approaching the COD, and ground contact time spent turning in the COD) by Musclelab 10.5.69 (Ergotest innovation A. S, Porsgrunn, Norway), which synchronized the laser gun and the contact mat. All the COD tests (including the sprint and braking test) were conducted on an indoor court surface (Taraflex Sport Evolution M 7.0 mm, Unisport, Finland).

### 2.4. Sprint Test

The sprint test consisted of a 30 m straight-line sprint. Athletes were instructed to complete the 30 m sprint within the shortest amount of time possible. The performance variable was the peak velocity, which was measured at 5, 10, 20, and 30 m using a laser gun. Step kinematics for the sprint (average step length, average step frequency, and average ground contact times) were also sampled.

### 2.5. Maximum Horizontal Braking Test

The maximum horizontal braking test was retrieved from a protocol by Harper et al. [37], using a standardized acceleration–deceleration test, included in the Musclelab v10.5.69 software (Ergotest Innovation A.S, Porsgrunn, Norway). Athletes were instructed to sprint 20 m with maximum effort, before initiating maximum deceleration after passing the 20 m mark (Figure 2). The laser was used to measure the deceleration after passing the 20 m mark. Furthermore, it was used to ensure that the athletes performed a maximal acceleration before deceleration of the laser-measured sprinting velocity at the 20 m marks. If it revealed a 5% decrease in velocity, compared to velocity after 20 m in the straight-line sprint, a reattempt was required after 3 min of rest. Horizontal acceleration power (W/kg), braking power (W/kg), and braking force (N/kg) were calculated from the braking test.

### 2.6. Strength Tests

The strength tests of the current study were similar to an earlier similar study by Falch, Rædergård, and van den Tillaar [29], but in men. It consisted of a bilateral back squat, a unilateral quarter squat performed on a Smith machine, and a lateral barbell squat. Performance was expressed as relative strength (load/body mass). Appropriate depth of the bilateral squat was defined as bending the knee until the trochanter major was in line with the patella. For the quarter and lateral squats, the athlete was required to bend the knee to a 90° flexion in the knee joint (Figure 3). The one-repetition maximum (1-RM) was estimated from the load–velocity relationship [38] using the best-fit regression line of three different data points for each individual athlete. Each data point represents load at a given velocity, whereby the average concentric velocity corresponds to ~1, 0.8, and 0.5 m/s. Thus, the different strength tests required maximal mobilization in the concentric phase of the lift, with three repetitions at each load. The average concentric velocity of the second and third repetitions of each series was used as a data point for calculating 1-RM. This is because, in lighter loads (<80% of 1-RM), the second or third repetition is often the fastest repetition [39]. A linear encoder sampling at 500 Hz (ET-Enc-02, Ergotest Technology AS, Porsgrunn, Norway) was used to measure the concentric velocity in the strength tests. All unilateral tests were performed with the right foot. Performance in the strength tests was estimated by 1-RM/body mass.

### 2.7. Plyometric Tests

The plyometric tests were also based on a protocol by Falch, Rædergård, and van den Tillaar [29] consisting of a unilateral vertical countermovement jump for maximal height (cm) and a unilateral lateral countermovement jump for maximal length (cm), herein referred to as a skate-jump and a drop jump with a drop height of 20 cm [40], whereby the reactive strength index (RSI) is the performance variable (jump height/ground contact time). In the unilateral vertical countermovement jump and the drop jump, athletes were instructed to place their hands akimbo, to prevent an arm swing from possibly contributing to performance, limiting the isolated effect of leg power [41]. In the vertical jump, the athlete was furthermore instructed to keep the passive foot in a locked position, to avoid the momentum of the passive foot from contribution to jump height. Reactive strength index (jump height/foot contact) in the drop jump and jump height in the unilateral countermovement jump were determined using a dual force plate (Ergotest Technology AS, Porsgrunn, Norway) sampling at 1000 Hz. The force plate registers contact time and flight time and calculates jump height with the use of flight time according to the following equation: $jump\;height=½\times 9.81\times {\left(\frac{flight\;time}{2}\right)}^{2}$.

All tests were performed with three approved attempts, with 3 min rest between each attempt. The average of all three attempts was used for further analysis.

### 2.8. Statistical Analysis

The descriptive statistics are presented as the mean ± standard deviation. To investigate the association between strength and plyometric capabilities with performance in force- and velocity-oriented CODs, correlations between the performance variables were determined utilizing Pearson’s correlational coefficient. The relationships between performance variables were based upon *r*-values defined as small (0.1 < *r* < 0.3), moderate (0.3 < *r* < 0.5), large (0.5 < *r* < 0.7), and very large (0.7 < *r* < 0.9) [42]. Utilizing the median split analysis based on the athlete’s average force and velocity COD performances, athletes were furthermore divided into fast and slow COD groups of eight athletes each. The median group (*n* = 9) was excluded from the statistical analysis. Comparisons of the two groups were conducted to investigate the study’s secondary objective if fast vs. slow COD performers possessed different step and acceleration/deceleration abilities. Group differences were investigated with multiple independent *t*-tests (fast vs. slow COD performers). The assumption of homogeneity of variance was controlled for with Levene’s test. The assumption of normality was confirmed for all variables using the Kolmogorov–Smirnov test. When assumptions for the independent *t*-test were violated, the Mann–Whitney U test was conducted. The effect of group differences is presented as Cohen’s *d*, whereby an effect of 0–2 constitutes a very small effect, 0.2–0.5 constitutes a small effect, 0.5–0.8 constitutes a moderate effect, and >0.8 constitutes a large effect. Furthermore, >1.2 was defined as a very large effect and >2 was defined as a huge effect [43]. The level of significance was set at *p* < 0.05, and the confidence interval was set at 95%. All tests were performed in SPSS v.27 (IBM Corp., Armonk, NY, USA).

## 3. Results

The mean performance in the different strength, plyometric, COD, sprint, and horizontal braking tests for all athletes (*n* = 25) is presented in Table 1.

### 3.1. Correlations

The bilateral and quarter squat revealed ’moderate to strong’ significant correlations with all total-time and first 10 m CODs (*r* > −0.43, *p* < 0.04, Table 2), except for the quarter squat and 20 m 180° COD, where the relationship was nonsignificant (*r* = −0.39, *p* > 0.07). The reactive strength index was only significantly moderately correlated with 10 m 180° COD (*r* = −0.44, *p* = 0.03), while all the other tests revealed insignificant small correlations with COD performance (*r* < 0.28, *p* > 0.35).

### 3.2. Fast vs. Slow Performers

When comparing the fast (*n* = 8) vs. slow COD performers (*n* = 8) in both the force- and the velocity-oriented CODs, as well as the different strength and plyometric tests, the fast performers were found to be significantly stronger in the bilateral and unilateral squat (F < 1.82, *d* > 1.35, *p* < 0.03). The fast performers also jumped significantly higher in the unilateral countermovement jump (F < 0.33, *d* > 1.29, *p* < 0.03, Table 3).

### 3.3. Step Kinematics Differences

The F-statistics from Levene’s test revealed unequal between-group variance for step frequency and flight time in the 30 m sprint and 180° COD (Table 4 and Table 5). When comparing fast vs. slow performers in the 30 m sprint and both CODs, a huge effect was observed for peak velocities across all the tests and distances, whereby the fast performers revealed significantly higher velocities compared to the slow performers (*d* > 3.31, *p* < 0.01). A large to huge effect was also observed for average ground contact time, flight time, and step frequencies in which the fast performers had shorter ground contact times, shorter flight times, and higher step frequencies (*d* > 1.26, *p* < 0.05). The fast performers also revealed greater horizontal acceleration and braking power in the horizontal braking test (*d* > 1.34, *p* < 0.02), where the effect was very large to huge (Table 4 and Table 5). Furthermore, a large nonsignificant effect was observed in the plant step and braking force between the fast and slow performers in the velocity-oriented COD, where the fast performers produced more braking force and shorter ground contact time in the plant step compared to the slow performers (*d* > 1.07, *p* > 0.07, Table 5).

## 4. Discussion

The primary objective of the current study was to examine the association between performance in strength and plyometric exercises and force- and velocity-oriented COD performance in young female court and field sport athletes. The correlational analysis revealed moderate and strong associations between relative strength in the bilateral squat and quarter squat with several of the COD performances (*r* ≥ −0.43, *p* ≤ 0.04). This result is in line with earlier research, indicating dynamic lower-limb strength to be important for COD performance in female athletes [44,45]. Furthermore, relative strength has been reported to be associated with COD performance [45,46,47].

Thus, the bilateral and quarter squat could share similarities with the acceleration/deceleration aspect prior to the plant step in both the force- and the velocity-oriented COD tests due to similar muscle requirements of knee-extensor and hip-flexor strength. Knee-extensor strength has been found to be associated with deceleration abilities [48], by eccentrically absorbing forces [36], which might allow higher velocities when initiating the COD movement. Furthermore, the quadriceps and gluteus maximus, which are agonist muscles in the bilateral and quarter squat, are suggested to contribute largely to acceleration moments when accelerating [49]. On the other hand, the lateral squat was the only strength exercise not revealing any significant correlation with any of the CODs (*r* = 0.28, *p* > 0.35), possibly due to the technical demands of the exercise regarding balance and control [50], inhibiting the athletes’ ability to maximize loads at the given velocities. This is logical, as training modalities often predominantly focus on movements in the sagittal plane [51], with exercises such as the bilateral and quarter squat. The balance requirements increase in unilateral movements and are further increased in free-weight exercises. Although it is a unilateral movement, instability in the unilateral quarter squat is reduced by being performed on a Smith machine [52].

Contradictory to earlier research [16,29,53,54], the correlation between plyometrics and COD performance was limited. Balance might also account for this finding, because instability may decrease the ability to express power [55,56]. The countermovement jump and skate jump are unilateral movements that were performed freely, demanding more balance than the unilateral quarter squat performed in the Smith machine when flexing the knee. As such, balance might inhibit a fast pre-stretch, which is desired for the muscles to reach a higher level of active state and subsequently aid in shortening velocities in the jumps [57,58]. Thus, the balance aspect in the eccentric pre-stretch of the jumps could negatively affect performance, which explains the ’small’ association with COD performance. Saeterbakken and Fimland [52] indicated that instability reduced the force output of the lower limbs, despite similar muscle activity. As such, it could be speculated that the reduced stability in the unilateral plyometric exercises reduces force output, limiting the correlation with COD.

The drop jump was the only bilateral plyometric exercise, whereby RSI was the only plyometric performance variable significantly correlated with COD performance, revealing a moderate association with the first 10 m 180° COD (*r* = −0.44, *p* = 0.03). This might be due to the relationship between RSI and deceleration ability because the first 10 m 180° COD is a force-oriented COD, whereby performance is more dictated by braking capabilities [22] compared to the 45° COD performances. Drop jumps and high-velocity decelerations share physical similarities, both demanding great eccentric strength and muscle activation in the gastrocnemius to absorb forces [29,36].

The observed correlations in the current study contradict earlier comparisons made in males [29], which suggested plyometric exercises to be more COD-specific. A possible explanation is that strength performance might reveal the greatest association with COD performance in populations with lower levels of relative strength. This is because there might be a threshold to how much strength is beneficial for developing physical abilities underpinning COD performance [59,60]. Furthermore, the bilateral and quarter squat were only significantly correlated with the first 10 m COD and total time to perform the COD test. The small to moderate observed correlation with the last 10 m CODs (*r* ≤ −0.35, *p* ≤ 0.1), accompanied by the distinctive magnitude of physical ability requirements in force- and velocity-oriented CODs [19,22], suggests the pivoting movement itself to be dependent on different neuromuscular abilities. The pivoting movement requires coordination of the whole body, such as the appropriate inclination angle of the trunk to manipulate the base of support and overcome inertia [21,61]. Without trunk stability and appropriate inclination angles of the trunk, net forces produced while reaccelerating will be limited. As such, transitioning from the weight-acceptance phase to the reacceleration phase might be heavily dependent upon technical demands, limiting the isolated effect of the lower limbs to produce force. Therefore, further investigation into the physical abilities of a priori reaccelerating, differentiating fast and slow performances in force- and velocity-oriented CODs, could increase insights into the associations between COD performance and strength and plyometric performance.

The fast performers in both force- and velocity-oriented CODs revealed similar physical capabilities, indicating that the force- and velocity-oriented CODs consisted of similar demands. The fast COD performers possessed higher levels of acceleration and deceleration abilities. Firstly, the data showed that the fast COD performers attained higher peak velocities in both the force- and the velocity-oriented CODs, as well as sprinting distances over 5, 10, 20, and 30 m, and greater horizontal acceleration power in the horizontal braking test. This finding was expected, as peak velocities have been observed to be associated with COD performance [47,53].

The higher observed acceleration was due to a higher step frequency, as fast performers performed both force- and velocity-oriented CODs with higher step frequencies (*d* > 2.04, *p* < 0.01), while the step length was similar. According to earlier research on sprinting [62], higher step frequencies were expected in the fast performers, as running speed is a result of step length × step frequency [34]. Step frequency, again, is a product of ground contact time and flight time, whereby shorter contact times and/or flight time results in a higher step frequency. Following earlier COD research [63], the ground contact time was observed to differentiate between the groups, whereby the fast performers revealed shorter ground contact times in both force- and velocity-oriented CODs (Table 4 and Table 5). Furthermore, the fast performers also revealed shorter flight time in the velocity-oriented COD (Table 4 and Table 5). The shorter ground contact and flight times in the fast COD performers accounted for a faster acceleration/deceleration phase. As the step length was the same between the groups, horizontal force production was not indicated to differentiate the fast vs. slow performers. The findings are supported by the horizontal braking test, whereby the fast COD performers revealed greater horizontal braking power, but not horizontal braking force. As the total force production was over the same distance (work), but in a shorter time (work/t), power was higher in the fast performers (Table 4 and Table 5). To accelerate/decelerate body weight, athletes need to exert net forces to the ground [21,24], and shorter contact times indicate faster production of net ground reaction forces to change momentum in the acceleration/deceleration phase, according to Newton’s laws of motion. The forces required to change the body’s momentum are dependent on mass × velocity, and the rate of change in momentum is dependent on the time over which forces are applied (force × time = mass × velocity) [36].

A higher production of power was visible in the strength tests in which the fast performers had a higher 1-RM, which was based on the load–velocity relationship. As such, fast COD performers could, with similar submaximal loads to slow performers, perform at a higher velocity in the different strength tests, thereby performing the same workload over a shorter time, producing more power and, therefore, a higher calculated 1-RM. The findings are in line with earlier research by Barr, Sheppard, Agar-Newman, and Newton [60] who found relative strength in lower-limb exercises to be associated with ground contact time and sprinting velocities (*r* = 0.47 to 0.71) in male athletes accustomed to strength training. Higher power production by the fast COD performers may also explain the higher jump height in the unilateral countermovement jump, as jump height is a result of velocity at takeoff (Table 3). However, the aforementioned balance requirements of the lateral squat and skate jump could have inhibited the fast athletes’ ability to express power in these exercises.

### Limitations

Strength performances were, for practical reasons, estimated with regression, which can differ from true 1-RM values. To limit differences between estimated and true 1-RM, lifts were performed without a pause to increase ecological validity and sampled from a relatively wide velocity range (±0.5 m/s) [64]. Furthermore, to save testing time and avoid fatigue, performance was only tested for the right foot in the unilateral tests, as the right foot performed the plant step in the COD tests, which does not account for lower-limb asymmetries. However, similar research has observed similar performance in force- and velocity-oriented CODs with a left and right turn [23]. Future research measuring lean body mass and muscle activity in the different tests is warranted, which could provide new useful insights into the nature of the results. Furthermore, testing “stronger” athlete groups would be important to investigate if the correlations for the plyometric and strength exercise with COD performance would change more in the direction of stronger correlations with plyometrics, as found in men [29].

## 5. Conclusions

In female court and field sport athletes, the bilateral and quarter squat revealed the greatest association with COD performance, possibly due to the demands of the knee-extensor and hip-flexors for rapidly applying force when accelerating/decelerating in a COD. Surprisingly, the association between plyometric and COD performance was limited. These results indicate that stronger athletes have higher levels of power production, positively influencing COD performance. As such, increases in lower-limb strength might positively influence COD performance if the acquired strength gains enhance the ability to produce more workload in a shorter time. Fast vs. slow performers in force- and velocity-oriented CODs were differentiated by higher peak velocities, as well as horizontal acceleration and braking power, which were observable through higher step frequencies and shorter ground contact times. The observed between-group differences in step kinematics might have been a result of differences in relative strength and ability to produce power.

### Practical Applications

Relative strength in the bilateral and quarter squat was found to be associated with both force- and velocity-oriented COD performance. According to the results of the current study, female team-sport athletes displaying relative strength of ~<1.5 load/BM in the bilateral back squat and ~<1 load/BM in the unilateral quarter squat might improve COD performance by increasing relative strength in these exercises. However, the practical application does not account for at what point more velocity-specific exercises, such as plyometrics, should be implemented.

## Figures and Tables

**Figure 1 jfmk-06-00083-f001:**
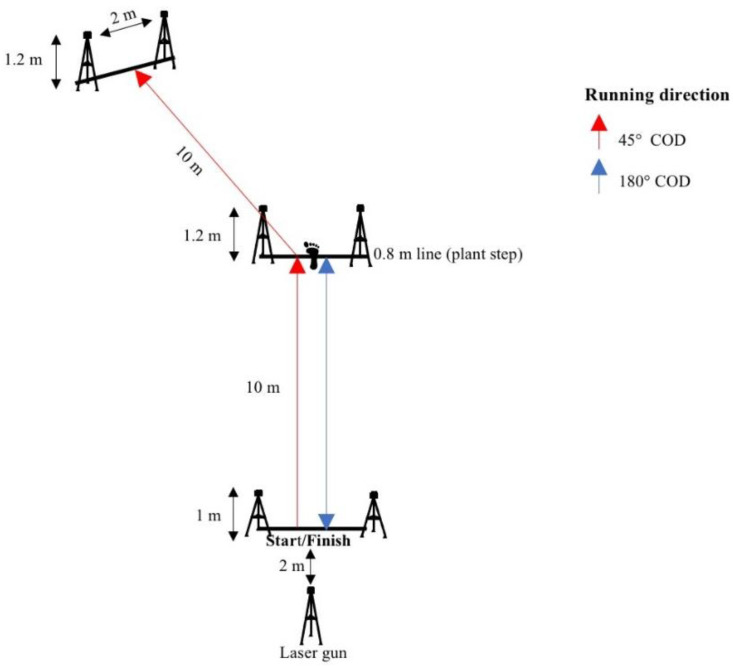
Setup for the COD track.

**Figure 2 jfmk-06-00083-f002:**
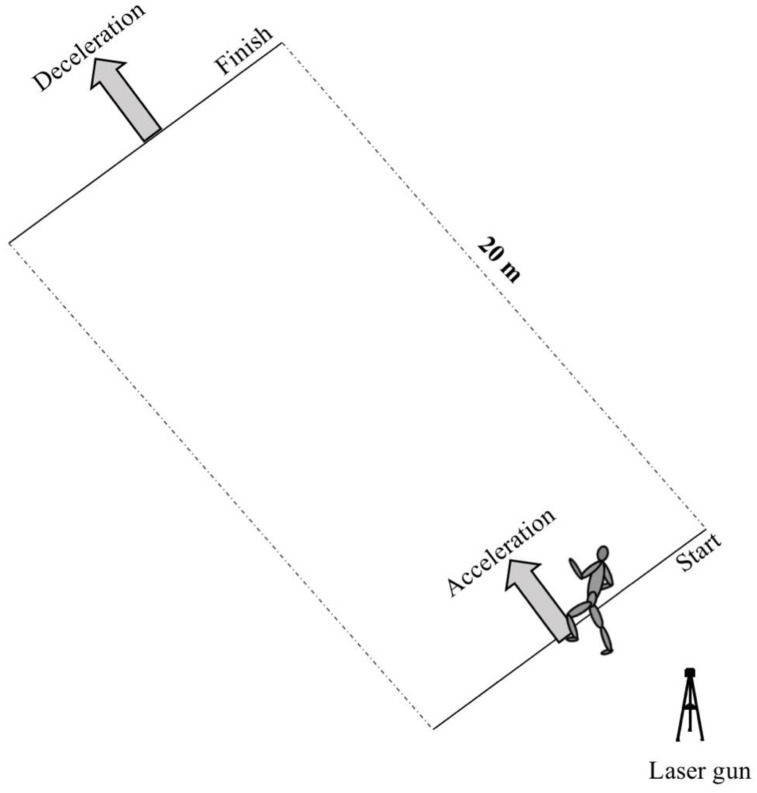
Setup for the maximum horizontal braking test.

**Figure 3 jfmk-06-00083-f003:**
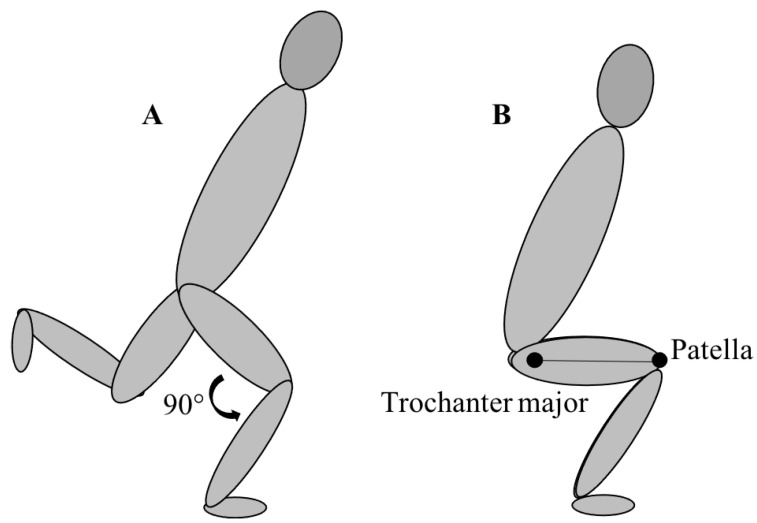
Depth requirements for the unilateral quarter squat, lateral squat (**A**), and bilateral back squat (**B**).

**Table 1 jfmk-06-00083-t001:** Descriptive statistics of performance in the different tests.

Strength Tests	Mean ± STD
Bilateral squat (1-RM/body mass)	1.28 ± 0.32
Unilateral quarter squat (1-RM/body mass)	0.88 ± 0.28
Lateral squat (1-RM/body mass)	0.57 ± 0.2
Plyometric tests	
Drop jump (reactive strength index)	117 ± 30.8
Unilateral countermovement jump (cm)	11.6 ± 2.5
Skate-jump (cm)	169.1 ± 9.7
CODs (s)	
First 10 m 45°	2.20 ± 0.11
Last 10 m 45°	1.68 ± 0.11
20 m 45°	3.88 ± 0.2
First 10 m 180°	2.40 ± 0.12
Last 10 m 180°	2.77 ± 0.17
20 m 180°	5.17 ± 0.24
Peak velocities in the straight-line sprint (m/s)	
5 m	5.36 ± 0.25
10 m	6.30 ± 0.32
20 m	6.89 ± 0.38
30 m	7.06 ± 0.48
Braking test	
Horizontal acceleration power (W/kg)	8.39 ± 1.22
Horizontal braking power (W/kg)	−10.51 ± 1.48
Horizontal braking force (N/kg)	−3.11 ± 0.47

STD = standard deviation; 1-RM = one-repetition maximum; COD = change of direction.

**Table 2 jfmk-06-00083-t002:** Correlation of performance in the different sprint, strength, plyometric, braking, and COD tests.

	Bilateral Squat	Quarter Squat	Lateral Squat	RSI	CMJ	Skate-Jump
COD first 10 m 180°	−0.46 *	−0.49 *	0.28	−0.44 *	−0.25	−0.13
COD last 10 m 180°	−0.35	−0.21	0.22	0.03	−0.3	0.15
COD 20 m 180°	−0.48 *	−0.39	0.28	−0.19	−0.28	0.05
COD first 10 m 45°	−0.64 *	−0.62 *	0.19	−0.2	−0.29	−0.01
COD last 10 m 45°	−0.29	−0.17	0.07	−0.02	−0.31	0.02
COD 20 m 45°	−0.5 *	−0.43 *	0.14	−0.13	−0.22	−0.05

CMJ is the unilateral countermovement jump; RSI is the reactive strength index; * indicates a significant correlation at the *p* < 0.05 level. COD = change of direction.

**Table 3 jfmk-06-00083-t003:** Differences in fast vs. slow performers in the different strength and plyometric exercises.

	Force-Oriented CODs		Velocity-Oriented CODs	
	Fast Performers	Slow Performers	Fast Performers	Slow Performers
**Strength (load/BM)**				
Bilateral squat	1.52 ± 0.31 *	1.05 ± 0.28	1.52 ± 0.23 *	1.1 ± 0.28
Quarter squat	1.06 ± 0.18 *	0.71 ± 0.16	1 ± 0.24 *	0.72 ± 0.17
Lateral squat	0.56 ± 0.17	0.61 ± 0.24	0.61 ± 0.28	0.56 ± 0.17
**Plyometric exercises**				
Drop jump (RSI)	123.8 ± 28.4	123.3 ± 40.8	112.9 ± 22.8	111 ± 35.9
CMJ (cm)	13.4 ± 1.4 *	10.1 ± 2.4	12.6 ± 1.4 *	10.3 ± 2.2
Skate-jump (cm)	165.5 ± 9.5	169.1 ± 8.3	167.8 ± 9.4	168.3 ± 8

CMJ is the unilateral countermovement jump (right); RSI is the reactive strength index; BM is the body mass; * indicates a significant between fast vs slow performers at the *p* < 0.05 level. COD = change of direction.

**Table 4 jfmk-06-00083-t004:** Differences in step peak velocities and step kinematics between the fast and slow performers in force-oriented COD.

Force-Oriented CODs						
	Fast COD Performers	Slow COD Performers	F	ES	ES	CI (95%)
	Mean ± STD	Mean ± STD		(*d*)	Description	
Peak velocities (m/s)						
First 5 m (of 30 m sprint)	5.61 ± 0.13	5.08 ± 0.16	0.22	3.61 *	Huge	−0.69, −0.37
First 10 m (of 30 m sprint)	6.62 ± 0.15	5.95 ± 0.2	0.85	3.76 *	Huge	−0.85, −0.47
First 20 m (of 30 m sprint)	7.26 ± 0.17	6.48 ± 0.24	0.7	3.78 *	Huge	−1.00, −0.56
30 m sprint	7.49 ± 0.23	6.56 ± 0.27	1.67	4.14 *	Huge	−1.19, −0.67
180° COD	5.77 ± 0.12	5.21 ± 0.18	2.72	3.78 *	Huge	−0.73, −0.39
Ground contact times (ms)						
30 m sprint, average	154.3 ± 17.9	175.3 ± 12.6	2.02	1.38 *	Very large	4.43, 37.65
180° COD, plant step	1239.7 ± 192.11	1058.7 ± 167.8	0.09	1.01	Large	−37.45, 12.37
180° COD, average	177.4 ± 14.6	202.4 ± 13.3	0.05	1.79 *	Very large	1.00, 39.87
Flight time (ms)						
30 m sprint, average	88.3 ± 5.4	103 ± 10.6	#	1.84 *	Very large	5.11, 24.26
180° COD, average	54.4 ± 9.5	67.6 ± 14.7	0.51	1.09	Large	−0.08, 26.42
Step lengths (m)						
30 m sprint	1.53 ± 0.1	1.52 ± 0.13	0.62	0.11	Very small	−0.14, 0.11
180° COD	1.19 ± 0.08	1.19 ± 0.07	0.64	0.1	Very small	−0.09, 0.08
Step frequencies (n/s)						
30 m sprint	4.22 ± 0.37	3.67 ± 0.14	#	2.18 *	Huge	−0.86, −0.24
180° COD	4.35 ± 0.34	3.74 ± 0.16	#	2.44 *	Huge	−0.91, −0.32
Horizontal braking test						
Acceleration power (W/kg)	9.26 ± 1.03	7.19 ± 0.89	0.15	2.15 *	Huge	−3.90, −1.35
Braking power (W/kg)	−11.58 ± 1.01	−9.73 ± 1.77	0.84	1.34 *	Very large	0.29, 3.38
Braking force (N/kg)	−2.88 ± 0.66	−3.34 ± 0.32	3.86	0.09	Very small	−0.09, 1.01

* Indicates a significant difference at the *p* < 0.05 level; # indicates violated assumptions and the conduction of the Mann–Whitney U test; F is the F-statistic from Levene’s test; ES is the effect size; *d* is Cohen’s *d*; CI is the confidence interval. STD = standard deviation. COD = change of direction.

**Table 5 jfmk-06-00083-t005:** Differences in performance-related measures between the fast and slow performers in velocity-oriented CODs.

Velocity-Oriented CODs						
	Fast COD Performers	Slow COD Performers	F	ES	ES	CI (95%)
	Mean ± STD	Mean ± STD		(*d*)	Description	
Peak velocities (m/s)						
First 5 m (of 30 m sprint)	5.58 ± 10	5.08 ± 0.16	2.96	3.79 *	Huge	−0.64, −0.35
First 10 m (of 30 m sprint)	6.59 ± 0.1	5.94 ± 0.18	3.34	4.69 *	Huge	−0.81, −0.50
First 20 m (of 30 m sprint)	7.25 ± 0.11	6.44 ± 0.18	3.86	5.48 *	Huge	−0.97, −0.65
30 m sprint	7.53 ± 0.14	6.55 ± 0.25	2.85	6.75 *	Huge	−1.13, −0.43
45° COD	6.01 ± 0.12	5.48 ± 0.2	3.7	3.31 *	Huge	−0.73, −0.36
Ground contact times (ms)						
30 m sprint, averages	156 ± 16.5	176.8 ± 10.1	2.5	1.49 *	Huge	5.20, 34.55
45° COD, plant step	168.2 ± 20.6	186.7 ± 12.3	1.55	1.12	Large	−0.07, 38.78
45° COD, averages	178.9 ± 14.5	195.9 ± 12.4	0.17	1.26 *	Very large	3.34, 35.46
Flight time (ms)						
30 m sprint, average	89.4 ± 8.9	102.3 ± 9.9	0.1	1.38 *	Very large	3.13, 26.5
45° COD, average	56.5 ± 12.4	97.1 ± 36.2	0.01	1.71 *	Very large	1.66, 39.97
Step lengths (m)						
30 m sprint	1.56 ± 0.1	1.54 ± 0.14	0.59	0.15	Very small	−0.15, 0.11
45° COD	1.18 ± 0.15	1.3 ± 0.14	0.55	0.9	Large	−0.04, 0.3
Step frequencies (n/s)						
30 m sprint	4.2 ± 0.35	3.66 ± 0.15	#	2.16 *	Huge	−0.85, −0.24
45° COD	4.33 ± 0.39	3.66 ± 0.26	0.63	2.04 *	Huge	−1.08, −0.25
Horizontal braking test						
Acceleration power (W/kg)	9.25 ± 1	7.21 ± 0.91	0.06	2.15 *	Huge	−3.07, −1.02
Braking power (W/kg)	–11.33 ± 0.91	–9.49 ± 1.65	1.06	1.44 *	Very large	0.42, 3.27
Braking force (N/kg)	–3.27 ± 0.25	–2.81 ± 0.61	4.59	1.07	Large	−0.04, 0.96

* Indicates a significant difference at the *p* < 0.05 level; # indicates violated assumptions and the conduction of the Mann–Whitney U test; F is the F-statistic from Levene’s test; ES is the effect size; *d* is Cohen’s *d*; CI is the confidence interval. STD = standard deviation. COD = change of direction.

## Data Availability

The raw data supporting the conclusions of this article will be made available by the authors, without undue reservation.

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
