# Peer review of "Association of Performance in Strength and Plyometric Tests with Change of Direction Performance in Young Female Team-Sport Athletes"

_jfmk, 2021, doi:10.3390/jfmk6040083_

Round 1
Reviewer 1 Report
Association of performance in strength and plyometric tests with change of direction performance in young female team-sport athletes
This is a well written study. Both research questions and designing are in the highest level. I comment the authors for the effort to measure athletes and especially female athletes. The study aimed to investigate the relationship between change of direction in two different conditions and strength, power performance. This is a real world study and provides useful information’s for coaches and scientists about the role of muscle strength and power in change of direction. Obviously, I have only minor comments to authors and a few suggestions that I would like to be added inside the manuscript.
Abstract:
Abstract is well written and provides a very good overview of the manuscript. Although it is analytical maybe adding the r-Pearson results from the correlational analysis would be more helpful for readers.
Introduction:
Introduction again is very analytical and steps on the literature. I agree with authors regarding the lack of studies in female athletes and this is highlighted inside the manuscript. May I suggest to authors to have a look to the following studies between strong and weak athletes in order to add a biological profile between these groups (Secomb et al., Int J Sports Physiol Perform. 2016;11(5);652-657; DOI: https://doi.org/10.1123/ijspp.2015-0481, Thomas et al., Int J Sports Physiol Perform. 2017;12;916-921; DOI: https://doi.org/10.1123/ijspp.2016-0317, Maden-Wilkinson et al., J Appl Physiol. 2020;128;1000–1011; DOI: https://doi.org/10.1152/japplphysiol.00224.2019). What I really miss here is the biological background which determines the difference between strong and weak athletes.
Methods:
Methods are very well described in the text. Authors use references for the protocols and the study can be replicated.
Subjects: I suggest to authors to change the word subjects and refer to the participants as athletes.
At which training period were the measurements performed? Were the athletes in a similar training status?
Procedures: Although the experimental procedures are clear I am not sure I understand the order of measurements and the duration of the experiment. Authors stated that there was a counterbalanced design but this would lead to at least 3 different visits to the laboratory for each athlete. I suggest to authors to add more details for the order of the measurements and the number of days needed to conclude the experiment.
Figures are perfect for readers. May I ask in the figure 3 how the athletes know when they reached the appropriate high of squat?
Skate jump: Is the skate jump the unilateral lateral countermovement jump for maximal length (cm)? Please, add this inside the Plyometric tests. Also, why authors performed only unilateral jumping tests and not bilateral as well (classic CMJ)? This is very interesting.
Results:
Results are great. I really enjoyed the tables. Not so strong correlations but still these data are novel for female athletes competing in real world conditions.
Discussion:
Discussion is very good. Here the results are clearly discussed and compared with other studies. Still, as mentioned above correlations between variables were moderate to almost large. Thus, I suggest to authors to add caution in the interpretation of these correlations. Still, much research is needed to reach certain conclusions.
Limitations: As mention above, some basic biological characteristics such as lean mass, muscle ultrasonography or EMG might have provided better insights into the nature of these results. Also, these findings were referred to female athletes (a strong aspect of the study) but still can these results be generalized to other athletic population?
Practical applications: This is a strong part of this study. The ratios added here are very helpful for coaches and trainers. Well done.
Author Response
Thank you for reviewing the paper. We have now changed the manuscript according to the comments of the reviewer.
point to point answering the comments:
Association of performance in strength and plyometric tests with change of direction performance in young female team-sport athletes
This is a well written study. Both research questions and designing are in the highest level. I comment the authors for the effort to measure athletes and especially female athletes. The study aimed to investigate the relationship between change of direction in two different conditions and strength, power performance. This is a real world study and provides useful information’s for coaches and scientists about the role of muscle strength and power in change of direction. Obviously, I have only minor comments to authors and a few suggestions that I would like to be added inside the manuscript.
Abstract:
Abstract is well written and provides a very good overview of the manuscript. Although it is analytical maybe adding the r-Pearson results from the correlational analysis would be more helpful for readers.
R-values have been added.
Introduction:
Introduction again is very analytical and steps on the literature. I agree with authors regarding the lack of studies in female athletes and this is highlighted inside the manuscript. May I suggest to authors to have a look to the following studies between strong and weak athletes in order to add a biological profile between these groups (Secomb et al., Int J Sports Physiol Perform. 2016;11(5);652-657; DOI: https://doi.org/10.1123/ijspp.2015-0481, Thomas et al., Int J Sports Physiol Perform. 2017;12;916-921; DOI: https://doi.org/10.1123/ijspp.2016-0317, Maden-Wilkinson et al., J Appl Physiol. 2020;128;1000–1011; DOI: https://doi.org/10.1152/japplphysiol.00224.2019). What I really miss here is the biological background which determines the difference between strong and weak athletes.
The authors appreciate the suggestion of the reviewer. However, the secondary objective of the current study was to investigate factors separating fast and slow athletes (such as strength), not what separates strong and weak athletes. We hope the reviewer understands our point of view.
Methods:
Methods are very well described in the text. Authors use references for the protocols and the study can be replicated.
Subjects: I suggest to authors to change the word subjects and refer to the participants as athletes.
“Subject” is now replaced with “athlete” throughout the manuscript, except for the heading and first sentence in 2.1.
At which training period were the measurements performed? Were the athletes in a similar training status?
Both training groups were tested “during the off-season” has now been added in the method section.
Procedures: Although the experimental procedures are clear I am not sure I understand the order of measurements and the duration of the experiment. Authors stated that there was a counterbalanced design but this would lead to at least 3 different visits to the laboratory for each athlete. I suggest to authors to add more details for the order of the measurements and the number of days needed to conclude the experiment.
The following sentence has been added in the method section: “Each group was randomly assigned to start with either the strength-, plyometric- or COD-tests and completed all the tests within the session.”
Figures are perfect for readers. May I ask in the figure 3 how the athletes know when they reached the appropriate high of squat?
Depth requirements was visually controlled for by the researchers.
Skate jump: Is the skate jump the unilateral lateral countermovement jump for maximal length (cm)? Please, add this inside the Plyometric tests. Also, why authors performed only unilateral jumping tests and not bilateral as well (classic CMJ)? This is very interesting.
Yes, skate-jump is the lateral countermovement jump for maximal length. This is now addressed.
The authors considered the bilateral CMJ as well, but the number of tests had to be limited to respect the athlete’s time. Thus, the bilateral CMJ was excluded as we already included a bilateral plyometric exercise (drop jump) and a vertical CMJ. The reasoning was to match the strength- and plyometric exercises in bilateral/unilateral exercises and direction of ground reaction forces produced.
Results:
Results are great. I really enjoyed the tables. Not so strong correlations but still these data are novel for female athletes competing in real world conditions.
Discussion:
Discussion is very good. Here the results are clearly discussed and compared with other studies. Still, as mentioned above correlations between variables were moderate to almost large. Thus, I suggest to authors to add caution in the interpretation of these correlations. Still, much research is needed to reach certain conclusions.
The authors agree with the reviewer’s perspective of interpreting correlations with caution. However, we believe that the correlations are already interpreted with caution. Is there a particular line the reviewer would like caution to be specifically mentioned?
Limitations: As mention above, some basic biological characteristics such as lean mass, muscle ultrasonography or EMG might have provided better insights into the nature of these results. Also, these findings were referred to female athletes (a strong aspect of the study) but still can these results be generalized to other athletic population?
The following sentence has been added: “The current study did not measure lean body mass, nor muscle activity in the different tests, which could provide useful insights into the nature of the results.”
The generalizability of the observations made in the current study to other athletic populations is only speculative. We do not have data to conclude that the observations are a result of gender- or strength-differences with reference to similar studies in men.
Practical applications: This is a strong part of this study. The ratios added here are very helpful for coaches and trainers. Well done.
Thank you
Reviewer 2 Report
Thank you to the authors for submitting their manuscript to International Journal of Environmental Research and Public Health; I enjoyed reading it.
There some suggestions that I think will provide clarity to the reader (outlined below).
Good luck with your amendments and I look forward to seeing the revised version.
SPECIFIC COMMENTS
Abstract:
Please include the kind of team sports (line 12)
Please add results of statistical analysis (line 17-21)
Keywords: “change of direction”, “strength” and “plyometric” are words included in the title. Please change it for another words
Introduction:
Please specify the sports of these female athletes (line 79).
Please change “won’t” to “will not” (line 84).
Material and Methods:
Please clarify the number of soccer and handball players. (line 114)
More information is needed about the subjects. Mean and SD body mass index, experience in football and others (line 115)
The season of this study should be provided (line 104)
Code number of Ethics Committee should be provided (122)
Did you forbid to consume caffeine? (line 125-126)
Please explain how you randomized the sample. (line 131)
Please add the only leg analysed (right) in CMJ test. Explain the reason and add references (line 140).
Please add the kind of boots/trainers used during tests (line 128-141)
Do you use wireless timing gate in sprint test? (line 177)
Results
Please explain in the table 1 footer: “COD”, “St. Dev”, “RM” (line 260)
Please explain in the table 2 footer: “COD” (line 261)
Please explain in the table 3 footer. The unilateral CMJ used (right), “COD” (line 278)
Please explain in the table 4 footer: “COD”, “STD” (line 297)
Where is in the table 4 “ES” and “CI”? I do not see it. (line 297)
Discussion:
It could be interesting to discuss the asymmetries between right and left leg and their relationship with COD test in the discussion section. Please consider the following references in female soccer players:
https://www.ncbi.nlm.nih.gov/pmc/articles/PMC8037528/
https://www.mdpi.com/2075-4663/7/1/29
https://pubmed.ncbi.nlm.nih.gov/31624414/
Are results of your study? Please clarify it. (line 316-318)
Please include a reference (line 354-356)
Please include future lines of future in the section “limitations” (line 417)
Author Response
Thank you for reviewing the paper. We have now changed the manuscript according to the comments of the reviewer.
point to point answering the comments:
Reviewer 2
Thank you to the authors for submitting their manuscript to International Journal of Environmental Research and Public Health; I enjoyed reading it.
There some suggestions that I think will provide clarity to the reader (outlined below).
Good luck with your amendments and I look forward to seeing the revised version.
SPECIFIC COMMENTS
Abstract:
Please include the kind of team sports (line 12)
“Handball and soccer players” has been added.
Please add results of statistical analysis (line 17-21)
Statistical results has been added.
Keywords: “change of direction”, “strength” and “plyometric” are words included in the title. Please change it for another words
Replaced with “power” and “step kinematics”
Introduction:
Please specify the sports of these female athletes (line 79).
Sentence has been revised: “This assumption is reasonable, as strength training has been found to enhance COD performance in both female volleyball athletes, and untrained females when measuring performance changes with the standardized T-test.”
Please change “won’t” to “will not” (line 84).
Changed.
Material and Methods:
Please clarify the number of soccer and handball players. (line 114)
N of the respective sport is now included.
More information is needed about the subjects. Mean and SD body mass index, experience in football and others (line 115)
Body mass index and minimum sessions per week is now included.
The season of this study should be provided (line 104)
The season of the study is now included.
Code number of Ethics Committee should be provided (122)
Project number is now provided.
Please explain how you randomized the sample. (line 131)
“by an online randomizer” has now been added.
Please add the only leg analysed (right) in CMJ test. Explain the reason and add references (line 140).
It is now explained that the tests were performed with the right leg, which performs the plant step in CODs with a left turn.
Please add the kind of boots/trainers used during tests (line 128-141)
The athletes wore boots designed for indoor court activities, not limited to a special type as the athletes prefer different types of boots. We do not feel this information is necessary but may add it if the reviewer disagrees with our point of view.
Do you use wireless timing gate in sprint test? (line 177)
Yes, timing gates was also placed at the different distances (5m, 10m, 15m, 20m and 30m) as a motivational tool. However, a laser gun is much more accurate than timing gates due to it measures accurately when moving at the start, while timing gates can be more inaccurate due to the arm and trunk movements in the start.
Results
Please explain in the table 1 footer: “COD”, “St. Dev”, “RM” (line 260)
Revised.
Please explain in the table 2 footer: “COD” (line 261)
Revised.
Please explain in the table 3 footer. The unilateral CMJ used (right), “COD” (line 278)
Revised.
Please explain in the table 4 footer: “COD”, “STD” (line 297)
Revised.
Where is in the table 4 “ES” and “CI”? I do not see it. (line 297)
This is now visible in the table.
Discussion:
It could be interesting to discuss the asymmetries between right and left leg and their relationship with COD test in the discussion section. Please consider the following references in female soccer players:
The authors appreciate the suggestion and agree that it would be interesting to discuss. However, the authors believe it would be beyond the scope of this article. We hope the reviewer understands our point of view.
https://www.ncbi.nlm.nih.gov/pmc/articles/PMC8037528/
https://www.mdpi.com/2075-4663/7/1/29
https://pubmed.ncbi.nlm.nih.gov/31624414/
Are results of your study? Please clarify it. (line 316-318)
The lines are based upon the results of the current study and earlier research. The lines provide a logical translation from the previous paragraph, which makes it difficult to revise these two lines only. However, we will revise the sentence/paragraph if the reviewer think it is needed.
Please include a reference (line 354-356)
The statement is introduced as a possible explanation whereby the following sentence provides rationale and references which the explanation is based upon.
Please include future lines of future in the section “limitations” (line 417)
The following sentences has been added: Future research measuring lean body mass and muscle activity in the different tests is warranted, which could provide new useful insights into the nature of the results. Furthermore, testing “stronger” athlete groups would be important to investigate if the correlations for the plyometric and strength exercise with COD performance would change more in the direction of stronger correlations with plyometrics as found in men.
Round 2
Reviewer 1 Report
No comments